# Effects of Temperature and Density on House Cricket Survival and Growth and on the Prevalence of Acheta Domesticus Densovirus

**DOI:** 10.3390/insects14070588

**Published:** 2023-06-29

**Authors:** Jozsef Takacs, Astrid Bryon, Annette B. Jensen, Joop J. A. van Loon, Vera I. D. Ros

**Affiliations:** 1Department of Plant and Environmental Sciences, University of Copenhagen, Thorvaldsensvej 40, DK-1871 Copenhagen, Denmark; jozsef.takacs@wur.nl (J.T.); abj@plen.ku.dk (A.B.J.); 2Laboratory of Entomology, Wageningen University and Research, Droevendaalsesteeg 1, 6708 PB Wageningen, The Netherlands; 3Laboratory of Virology, Wageningen University and Research, Droevendaalsesteeg 1, 6708 PB Wageningen, The Netherlands

**Keywords:** *Acheta domesticus*, house cricket, insect production, insects as food and feed, cricket viruses, entomopathogenic viruses, mass rearing

## Abstract

**Simple Summary:**

The growing world population demands an increase in food supply. Mass-rearing of insects can contribute to food production and recycling of nutrients. One of the insects that has a high potential as an alternative source of protein and other nutrients is the house cricket. However, cricket production faces various disease challenges, such as the Acheta domesticus densovirus (AdDV). This virus can cause high mortality in cricket colonies, and the triggers of virus outbreaks are not yet understood. Rearing crickets at lower densities and an optimal temperature may prevent viral disease outbreaks, but no studies examined the correlation between viral abundance and rearing densities and/or temperature. Therefore, this study examined the effect of different temperatures and rearing densities on cricket mortality and biomass and on the abundance of AdDV. In total, nine combinations of temperature (25, 30, 35 °C) and density (10, 20, 40 crickets) were tested. Higher rearing densities and temperatures resulted in higher total biomass produced per rearing unit and a minor impact on mortality. The results indicate that high rearing density can increase AdDV abundance, and that viral abundance is reduced at 35 °C.

**Abstract:**

The house cricket, *Acheta domesticus*, is a commonly reared insect for food and feed purposes. In 1977, a report described a colony collapse, which was caused by the single-stranded DNA virus Acheta domesticus densovirus (AdDV). Currently, there are no confirmed *A. domesticus* colonies free of AdDV, and viral disease outbreaks are a continuous threat to *A. domesticus* mass rearing. Correlations between cricket rearing density or temperature and AdDV abundance have been hypothesized, but experimental evidence is lacking. Optimised rearing conditions, including temperature and density, are key to cost-effective cricket production. In this study, house crickets were subjected to different combinations of rearing density (10, 20, 40 crickets per box) and temperature (25, 30, 35 °C) to study the effect on cricket survival, biomass, and AdDV abundance. Rearing temperature affected had a minor effect on survival, which ranged between 80 and 83%. Total cricket biomass increased with higher temperatures and higher densities. Viral abundance in crickets at the end of the rearing period was variable; however, high rearing density seemed to result in higher AdDV abundance. At 35 °C, a temperature considered suboptimal for house cricket production, viral abundance tended to be lower than at 25 or 30 °C.

## 1. Introduction

The continuing increase in the global human population urges for better use and distribution of resources, such as food, feed, and raw materials produced by agriculture [1]. Mass rearing of insects is proposed as a solution for better utilization of organic matter from the side and residual streams required to produce a sufficient amount of good quality food and feed [2]. Over the last decade, several companies started industrial-scale insect rearing to produce insects for food and feed purposes [3]. The insect-rearing industry aims to utilize and upcycle (low) value organic resources [4], thereby producing alternatives to the current protein sources with a relatively lower environmental impact [5].

For several decades, the house cricket *Acheta domesticus* L. has been reared as a pet/zoo feed in Europe and North America and as a food source in Asia [6,7]. This cricket has a favourable nutritional profile [8,9], which allows its direct utilization by the food industry as an ingredient of various products [10,11].

However, the mass production of this species can be easily jeopardized by the Acheta domesticus densovirus (AdDV), of which the first report dates back to 1977, describing a colony collapse due to a disease caused by this virus [12]. AdDV can cause massive disease outbreaks in *A. domesticus* colonies, causing high mortality and the loss of almost entire production batches [12,13,14,15,16]. At present, there are no confirmed virus-free *A. domesticus* colonies known. AdDV is a single-stranded DNA virus from the insect-specific *Densovirinae* subfamily of the *Parvoviridae* family [14]. Viruses from this subfamily are known to be highly virulent for their host organisms [17]. AdDV also infects other cricket species, but in these species, the infection does not result in mortality [15].

Cricket-rearing facilities are conducive for virus transmission due to the high degree of crowding, the high humidity, and the high temperature, conditions set to promote biomass output [18]. However, it is not yet understood if AdDV is mainly present in a covert state, in which the symptoms are not clearly visible and may include mild symptoms such as diminished feeding. Currently, there is no information on how the virus can switch from a covert state into an active, overt state, with apparently paralyzed and swollen crickets. The literature available on AdDV mainly covers its detection and quantification [12,19], but studies examining the transmission between individuals and different generations of crickets are lacking. During the growing phase of the crickets, several possible stress factors can be identified, such as non-optimal temperature, too high humidity, high rearing densities, or the presence of various entomopathogens [20]. These factors are hypothesized to interact with both each other and the host organism, making them possible triggers for symptomatic AdDV outbreaks.

Temperature is a crucial factor in insect production systems since insects are poikilothermic organisms, meaning their body temperature is dependent on the ambient temperature [21]. Insect developmental rate depends on temperature according to an optimum curve delimited by a minimum and maximum [22,23]; for house crickets, the optimal range is between 28 °C and 30 °C [24]. When the temperature in the rearing facility deviates from the optimal range, it can result in temperature stress, which can lead to negative physiological effects [25]. 

The infrastructure required for heating and cooling in rearing facilities is one of the major costs involved in both the construction and the operation of an insect production facility [26]. To achieve maximal efficiency in the production unit, it is, therefore, logical to aim for the highest possible rearing densities in order to increase productivity per surface area. In most cases, growers use egg trays and paper corridors to increase the surface area within the production unit and to provide dark hiding areas as a resting space for the crickets [27]. During the growth phase, house crickets increase in biomass from approximately 0.7 mg as pinhead (freshly hatched nymphs) to 300–500 mg of adult weight, an increase of ~500–750-fold (Jozsef Takacs, personal observations). Consequently, with the increase in body volume, the resting space will become smaller, and crowding can be observed in the production units. 

Optimal densities combined with optimal growth temperatures resulting in the highest yields per batch are favourable from a commercial production perspective. Unlike other mass-reared insect species [28,29,30,31], only a few studies investigated optimal rearing densities and temperatures for house crickets [28,32]. To date, it is unknown if there is a correlation between rearing density, temperature, and the abundance of the AdDV. Optimising rearing conditions for the best possible yields while maintaining low disease incidence is crucial for the economic viability of cricket mass production. Therefore, we investigated the possible consequences of different temperatures and densities on survival, biomass production, and AdDV abundance.

## 2. Materials and Methods

### 2.1. Cricket Rearing and Experimental Units

The house cricket colony kept at the Laboratory of Entomology, Wageningen University since 2006 was reared in plastic containers (40 L, containing ca. 500 adults) in a climate-controlled room at ~28 °C [33] and crickets were provided with chicken feed (Kuiken Opfokmeel, Kasper Fauna Food, Woerden, The Netherlands), carrot slices and water from water dispensers on a daily basis. For the experiment, 20-day-old crickets (calculated from the date of hatching) were randomly selected from the main rearing. The experimental units consisted of plastic boxes with a volume of 900 cm^3^ (10 × 15 × 6 cm) in which 10, 20, or 40 crickets were housed per box. Each box received a piece of cardboard egg tray (5 × 5 × 7 cm), water gel (DCM Aquaperla^®^, De Ceuster Meststoffen N.V., Grobbendonk, Belgium), and feed trays (3.5 × 0.2 cm) to provide the crickets with hiding space, water, and feed sources, respectively. In the 10 and 20 crickets per box densities, one feed tray was used, while in the 40 crickets per box density, two feed trays were used to avoid the possible stress caused by the lower individual-to-feeding surface ratio Expressed as the number of individuals per unit of volume, the lowest of the three densities we tested corresponded with the density commonly used in house cricket lab colonies, including the colony from which our experimental crickets originated. Our experimental design was aimed at testing density as a possible stress factor; hence we tested 2× and 4× higher densities. The boxes were observed daily, and water and feed were replenished when required to allow *ad libitum* drinking and feeding. The boxes were kept in incubators set to the corresponding treatment temperature of 25, 30, or 35 °C, all at 12:12 L:D period and on average 75–85% relative humidity. 

The experiments were finished when the first chirp sounds appeared, signalling male maturity. This was at varying time points due to the different temperatures resulting in different developmental durations (15–16 days at 35 °C, 21–22 days at 30 °C, and 30–31 days at 25 °C). At the end of the experiments, the number of crickets per cage was counted and considered as survivors; missing crickets were considered dead. When crickets were observed dead, they were not removed.

Three different temperatures combined with three different densities were tested, creating nine different treatments, with five replicate boxes per treatment. The experiment was serially repeated three times (Experimental runs 1, 2, and 3); thus, the total number of crickets used in the experiments was 3150. The three experimental runs were carried out between May and November 2022.

### 2.2. DNA Extraction and qPCR Analysis

At the end of the experiments, five adult crickets (three females and two males) were taken from every experimental unit (box) and stored at −20 °C until DNA extraction. At the start of each experimental run, cricket samples were also taken from the main rearing to determine the initial viral load since a negative control with virus-free crickets was not available. Homogenization of the crickets sampled was conducted by pestle and mortar with the addition of 0.5 mL of phosphate-buffered saline (PBS) solution as the first step of the DNA extraction. The homogenized samples were centrifuged for 15 min at 8000 RPM, after which the supernatants were removed and used for DNA extraction. The DNA extraction was carried out using the Qiagen dNeasy^®^ Blood and Tissue kit (Qiagen, Hilden, Germany), following the Total insect DNA extraction protocol, except for sample homogenization, which was conducted as described above. Quality control of the DNA samples was carried out using Nanodrop^®^ ND1000 Spectrophotometer (Thermo Fisher Scientific Inc., Waltham, MA, USA). 

Relative quantitative PCR (qPCR) was performed to quantify the number of AdDV viral copies in the samples. The *Acheta domesticus* elongation factor 1 alpha gene, partial coding sequence (Genbank access. nr. EU414685.2) with an amplicon size of 199 base pairs was used as host reference gene (see Table 1 for primer sequences). For the primer design, Primer3 (version 4.1.0) was used [34]. The AdDV primers targeted a gene encoding for a non-structural protein with the amplicon size of 96 base pairs and were adopted from the publication of Duffield et al. [18] (Table 1). Reactions were performed with SYBR™ Select Master Mix (Thermo Fisher Scientific Inc., Waltham, MA, USA) with a total reaction volume of 20 μL containing 3 μL template DNA, 5.4 μL autoclaved water, 10 μL SYBR™ Select Master Mix, 0.8 μL forward and 0.4 µM reverse primers. The protocol for the thermal cycling was as follows: UDG activation for 2 min at 50 °C and denaturation for 2 min at 94 °C, followed by 40 cycles of 30 s at 95 °C, 45 s at 59 °C and 1 min at 72 °C. Plate reads to measure fluorescence were conducted after every extension step. A visual inspection of the melting curve with a temperature gradient of 0.5 °C/5 s for the interval of 55 to 95 °C was carried out as quality control to ensure that only specific products were amplified [35]. Primer efficiencies were calculated by incorporating a standard curve for every plate individually and were in the range of 95–105% for both the host and the viral primers. The amplification was carried out using a CFX96 Touch Real-Time PCR Detection System and the CFX Maestro 5.3 software, both from Bio-Rad Laboratories, Inc., Hercules, CA, USA. The threshold cycle (Ct) for each well was determined by the CFX Maestro software using a single CQ (Ct) threshold determination mode. 

Raw abundance data from the qPCR runs were analysed using the qbase+ Version: 3.2 (Biogazelle, Zwijnaarde, Belgium—www.qbaseplus.com, accessed on 5 July 2022) software, with target and run specific amplification efficiencies. The abundance of AdDV was expressed as Calibrated Normalized Relative Quantities (CNRQ) [36]. This is achieved by first normalizing the CQ values to the host reference target (the host gene) and then by scaling the fold of change to the reference group of the start values (samples taken at the start of the experiments to determine initial viral abundance). After a logarithmic transformation, the CNRQ values were exported for further analyses in R Studio.

### 2.3. Data Analyses and Visualization

Data were analysed using R, version 4.2.2 (31 October 2022) [37] and R Studio Team (2020).

To test if the data sets were normally distributed, a Shapiro–Wilk test of normality was used [38]. If the data met the assumptions of normality and homogeneity of variance, analysis of variance (ANOVA) was used to test the differences between the treatments. The following variable met the assumption of normality: relative abundance data.

If the data was not normally distributed, generalized mixed effect models (GLM) were built using the lme4 package of R [39]. The following variables did not meet the assumption of normality: cricket survival, total biomass, and female and male individual weight. The GLM models were compared based on the Second-order Akaike information criterion (AIC) values, and the best-fitting models with the lowest AIC were selected [40]. In the case of the total biomass, female and male individual weight data, the model including an interaction between temperature and density was the model best fitting the data; the experimental run included a random factor. A GLM with binomial distribution was used to compare survival. The best-fitting model for the survival data used temperature as a fixed factor and the experimental run as a random factor. 

In case a factor had a significant effect, pairwise comparisons were carried out with the Package ‘emmeans,’ version 1.8.4-1 from R [41] or with Tukey’s honest significant test (HSD) test. Data were visualized using the R package ggplot2 [42].

## 3. Results

The following experiments combined three densities and three temperatures into nine treatments to test their effect on cricket survival, total biomass, and relative viral abundance.

### 3.1. Cricket Survival

The temperature had a main effect on cricket survival (Figure 1, Appendix A). No effect of density or interaction between temperature and density was detected on survival. Survival was higher for crickets reared at 25 °C than for those reared at 30 °C (*p* = 0.0111). No difference in cricket survival was seen between 25 °C and 35 °C or 30 °C and 35 °C (*p* = 0.5210 and 0.1693, respectively). 

### 3.2. Cricket Biomass

#### 3.2.1. Total Biomass of the Surviving Individuals

The total harvested biomass of surviving crickets differed significantly between treatments; there was an interaction between rearing temperature and rearing density (Figure 2, Appendix A). The three treatments with the lowest rearing density of 10 individuals per box resulted in significantly lower produced biomass compared to all other treatments. Between the three treatments with density 10, at 25 °C, a significantly lower weight was produced compared to the 30 °C and 35 °C treatments (*p* = 0.0002 and 0.0004, respectively). There was no significant difference at density 10 between the temperatures 30 °C and 35 °C (*p* = 1.000). At the density of 20 individuals per box, the pattern was similar to density 10; at the two higher temperatures of 30 °C and 35 °C, significantly higher biomass was recorded than at 25°C (*p* = 0.0098 and 0.0375, respectively). The biomass at 30 °C and 35 °C for density 20 did not differ significantly from each other (*p* = 1.000). Among all treatments, those with a density of 40 individuals per box resulted in the highest average total harvested biomass. However, total biomass at the density of 40 individuals per box was similar at the three tested temperatures (*p* = 0.1112, 0.4317, and 0.9988, respectively). The exact *p* values for the pairwise comparisons can be found in Appendix A. 

#### 3.2.2. Individual Weight of the Surviving Crickets, Separated by Genders

The median individual weight of the crickets was significantly affected by both temperatures and by density. Since gender had a significant effect on the individual weight (*p* < 0.0001), with females being heavier than males, the individual weight data for the two genders were analysed separately. The median individual weight of the surviving female crickets differed significantly between treatments (Figure 3 and Appendix A), showing an interaction between rearing temperature and rearing density. The lowest weight was achieved at 25 °C, which was significantly lower than 30 °C and 35 °C. The female individual weight was significantly higher at 30 °C compared to 35 °C, at the densities of 20 and 40 crickets.

The individual weight of the surviving male crickets differed significantly between treatments (Figure 4 and Appendix A); there was an interaction between rearing temperature and rearing density. Similarly to females, the median male individual weight was the lowest at 25 °C for all densities. Male individual weight differed significantly between 30 °C and 35 °C for the densities of 20 and 40. At 35 °C, the individual weight of males was significantly higher at density 10 compared to 40.

### 3.3. Relative Abundance of AdDV

The variance analysis showed a major, significant batch effect on the viral abundance data collected at the end of the experimental runs, as confirmed by a one-way ANOVA (*p*= 2 × 10^−16). Therefore, the three experimental runs were analysed separately. The initial viral load of the three batches of nymphs was similar (ANOVA, *p* = 0.773). There was no significant interaction effect between temperature and density on the relative viral abundance for any of the three experimental runs (*p* = 0.1635; 0.0866 and 0.5902, respectively) (Suppelementary Appendix A). In the case of experimental run 1, only temperature had a significant effect on the viral levels (*p* = 0.0405), being higher at 25 °C than at 35 °C; however, no effect of density was observed (*p* = 0.7845). For experimental run 2, neither an effect of temperature nor density was observed (*p* = 0.8391 and 0.2092, respectively). However, in experimental run 3, both temperature and density significantly affected viral abundance (*p* = 0.0024 and 0.0047, respectively). Among the three experimental runs (Figure 5), only in the first run a high fold of change in viral abundance was observed; the second and third runs resulted in minimal overall changes. In the case of experimental runs 1 and 3, rearing at 35 °C tended to result in lower relative viral abundance.

## 4. Discussion

For cold-blooded animals and especially insects, the environmental temperature has a strong impact on survival, developmental rate, reproduction, and behaviour [43,44,45]. The ambient temperature can be relatively easily manipulated within the cricket-rearing facilities using ventilation systems. To decide on an optimal temperature, it is advised to test a range of potential temperatures; however, this did not receive focus from research so far. 

In this study, the temperatures of 25 °C, 30 °C, and 35 °C were selected to cover the likely range of cricket production in a semi- or fully industrialized facility. Significant differences were observed for the biomass produced for the different rearing temperatures and densities tested. Harvested cricket biomass showed a minor effect of rearing at temperatures 30 °C and 35 °C with densities of 10 and 20 crickets per box. At the rearing density of 40 crickets per box, there were no differences in total biomass between the three temperatures tested. Another important aspect is the individual body weight of the crickets. Minor differences in individual body weight between the three densities tested were found. The temperatures of 30 °C and 35 °C resulted in higher body weight for both females and males. A higher individual body weight is profitable for producers. In addition, higher female body weight can be important for the breeding stock of an insect-rearing facility, as it will result in higher egg yields [46,47]. A recent study from 2023 showed that higher rearing densities could reduce the survival of house crickets significantly [32]. From the productivity perspective, the question might be raised: are the losses due to mortality caused by high rearing densities outweighed by the higher cricket biomass yield? In the results presented in this article, the increase of cricket rearing density interacted with the tested rearing temperatures, resulting in a clear increase in the total harvested cricket biomass. Further research should address the maximum rearing density that prevents competition for feed and space that may lead to reduced biomass harvest and to an increased hazard of disease epidemics since high densities are conducive for disease epidemics [48,49]. 

The economic impact of viral disease outbreaks is well known for several mass-reared insect species [50,51], and there are also references directly connecting densovirus outbreaks with economic losses in cricket production facilities [13]. Decreased rearing densities are hypothesized as a possible solution in house cricket rearing to avoid viral disease outbreaks, but experimental data supporting this practice is lacking. The viral abundance data of the experiments described in this article did not unequivocally confirm the hypothesis of higher rearing densities resulting in higher viral abundance over the density range studied, although a significantly higher abundance was found at density 40 in one of the three batches. However, this observation could be due to the high variation of the final viral loads between the experimental runs. High variation of viral levels is prevalent in the rearing, not just over time but also over life stages simultaneously present in the facilities. Insect densities in mass-rearing of house crickets are poorly documented in the literature; expressions are found as number of individuals per volume unit of rearing container or per surface unit [32] which complicates comparisons among studies. Moreover, in practice, the surface area in rearing containers is often substantially enlarged by adding carton material, e.g., carton previously used for holding chicken eggs, as applied in our study. Extrapolation of the densities we studied to mass-rearing scale is not straightforward and requires additional experiments. Additionally, our data showed that viral abundance was not impacted by density, but mainly by rearing temperature. Similarly, there was no effect of rearing density on survival (despite high relative viral loads in some cases), but the rearing temperature of 30 °C resulted in slightly lower survival than a temperature of 25 °C.

With increasing rearing densities, the occurrence of cannibalism should also be monitored. In our density experiments, we did not record if cannibalism occurred, but insect rearing often has to deal with losses caused by cannibalism when the rearing densities are high [31,52,53], and crickets are known to be highly cannibalistic [54]. Cannibalism can also make the insects more vulnerable to opportunistic pathogens since the cuticle, the natural barrier preventing the entry of the pathogens, is penetrated. This way, the first defence of the crickets can be bypassed by pathogens [55]. For densoviruses, however, this might be different since those infect mainly via oral ingestion [56]. To date, no research addressed the correlation between the rearing densities and the incidences of cannibalism in crickets. In the rearing of both Copenhagen University and Wageningen University and Research, personal observations were made of crickets cannibalising on asymptomatic and AdDV symptomatic individuals. When consuming symptomatic crickets, relatively high viral exposure can occur for the cannibalistic crickets, which can lead to a viral infection. However, cannibalism can also suggest feed depletion or may indicate that nutrients are lacking from the diet used in the rearing system [57,58,59], which can induce healthy individuals to cannibalise (diseased) individuals. This can initiate a chain reaction in the rearing, which might lead to the rapid colony collapses reported as a result of AdDV outbreaks. Lacking nutrients can also result in decreased immune functions of insects [60,61], making them less resistant to possible pathogen challenges. House crickets are commonly reared on chicken feed [62,63], but optimised diets should be developed to increase feed conversion efficiency and cost-effectiveness. These studies could also evaluate the effect of different diets on the abundance of AdDV. Furthermore, there are several other factors influencing the crickets during their rearing, such as humidity and light: dark cycles. The presence of other viruses and their interaction with the crickets and with AdDV can also have an impact on AdDV levels. Evaluating these parameters individually and in combination will resolve their effects on cricket production. 

## 5. Conclusions

Temperature and density are key parameters for the cost-efficiency of most animal production facilities. Our experiments aimed at evaluating the possible effects of varying rearing densities and temperatures on survival, total biomass, and AdDV abundance in house cricket rearing. Currently, AdDV is the main threat to house cricket rearing. Whereas a temperature of 35 °C that is considered suboptimal for house cricket rearing does not appear to be a trigger for AdDV outbreak, the highest density tested may trigger higher viral abundance, depending on an unknown factor favouring viral replication. The results presented in this article can be considered as a basis for further research aiming to optimise the production conditions of cricket rearing for maximum efficiency. Based on the economic aspects of the production, producers can evaluate what is more beneficial for their facility and rearing setup. However, the actual cost of heating is mainly dependent on the location and heating, ventilation, and air conditioning systems. Cricket-rearing facilities can be found in all climatic conditions; the local circumstances and costs will determine which temperatures are better suited for the production and/or are more economically viable to maintain in the facility. 

## Figures and Tables

**Figure 1 insects-14-00588-f001:**
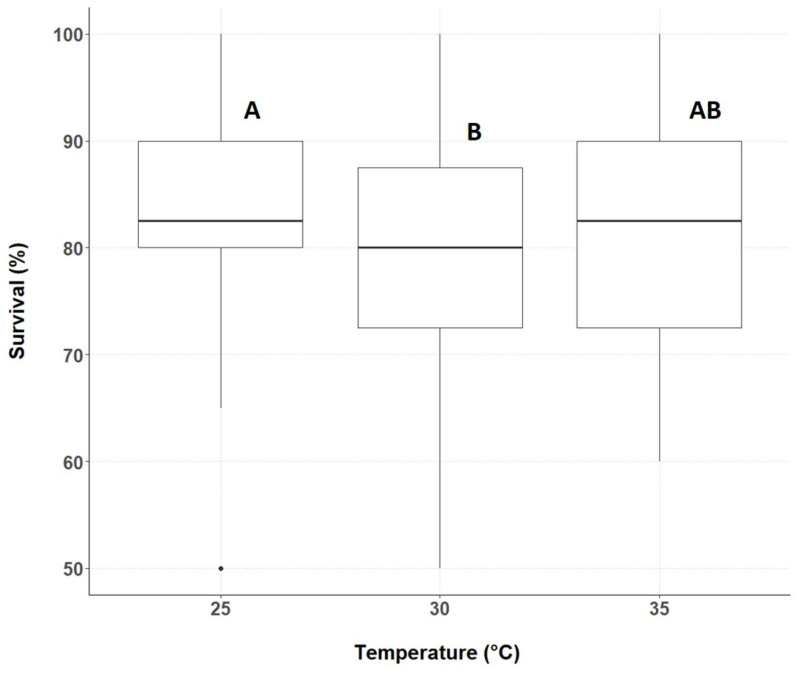
Median cricket survival was calculated over the three experimental runs. Temperatures tested are indicated along the X-axis. The Y-axis represents survival (%), horizontal lines depict the median, boxes indicate the lower and upper quartiles, vertical lines extend to minimum and maximum values, black dots represent outliers. Different letters indicate significant differences between the treatments.

**Figure 2 insects-14-00588-f002:**
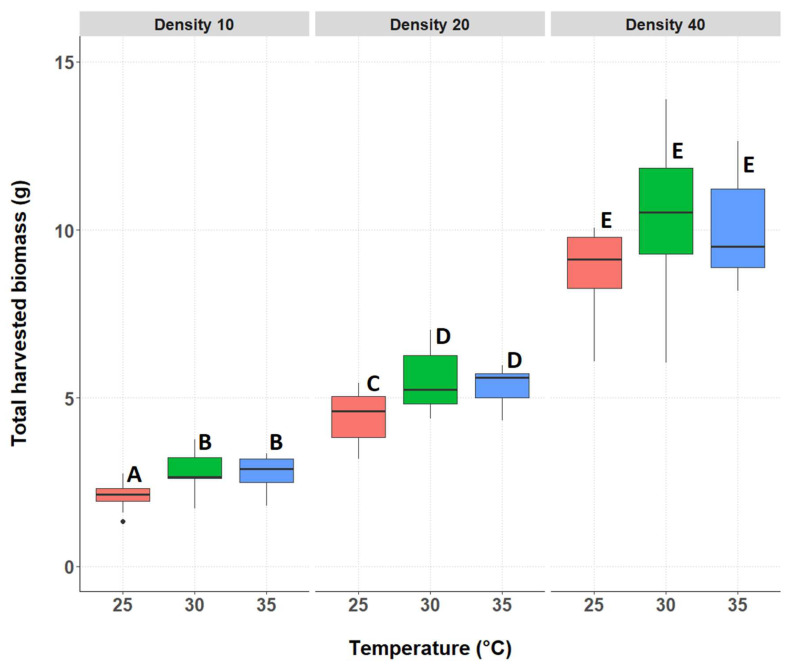
Total harvested biomass (g) for each treatment group. Temperatures are indicated along the X-axis per density tested, indicated above the figure (D10, D20, or D40). The boxplots show the median total biomass (bold horizontal line), boxes depict lower and upper quartiles, vertical lines extend to minimum and maximum values, black dots represent outliers. Different letters indicate significant differences between the treatments.

**Figure 3 insects-14-00588-f003:**
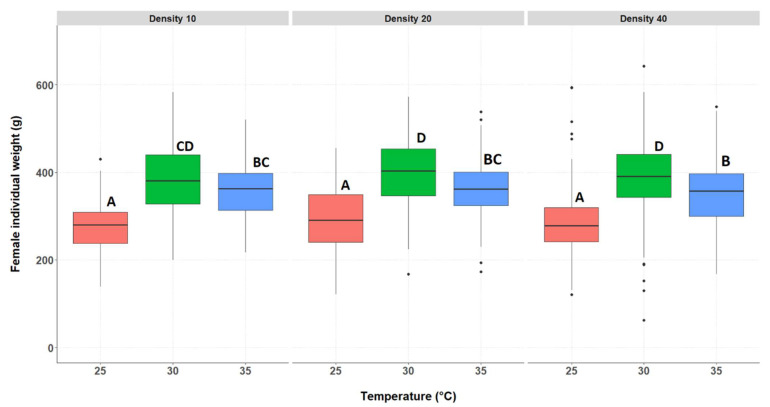
Female individual weight (g) for each treatment group. Temperatures are indicated along the X-axis per density tested, indicated above the figure (D10, D20, or D40). The boxplots show the median individual weight (bold horizontal line), boxes depict lower and upper quartiles, and vertical lines extend to minimum and maximum values, black dots represent outliers. Different letters indicate significant differences.

**Figure 4 insects-14-00588-f004:**
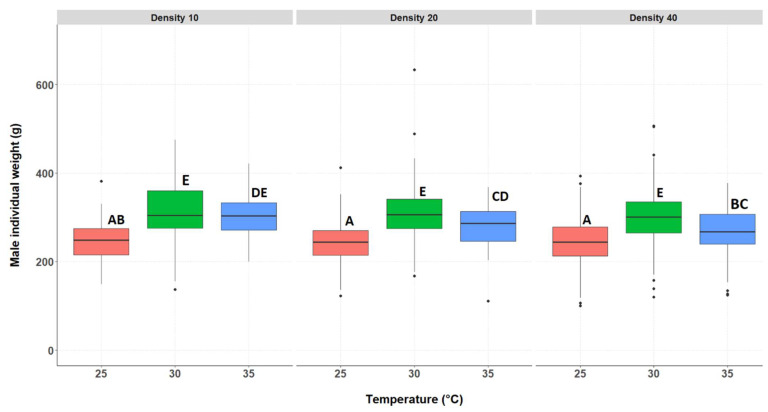
Male individual weight (g) for each treatment group. Temperatures are indicated along the X-axis per density tested, indicated above the figure (D10, D20, or D40). The boxplots show the median individual weight (bold horizontal line), boxes depict lower and upper quartiles, and vertical lines extend to minimum and maximum values, black dots represent outliers. Different letters indicate significant differences.

**Figure 5 insects-14-00588-f005:**
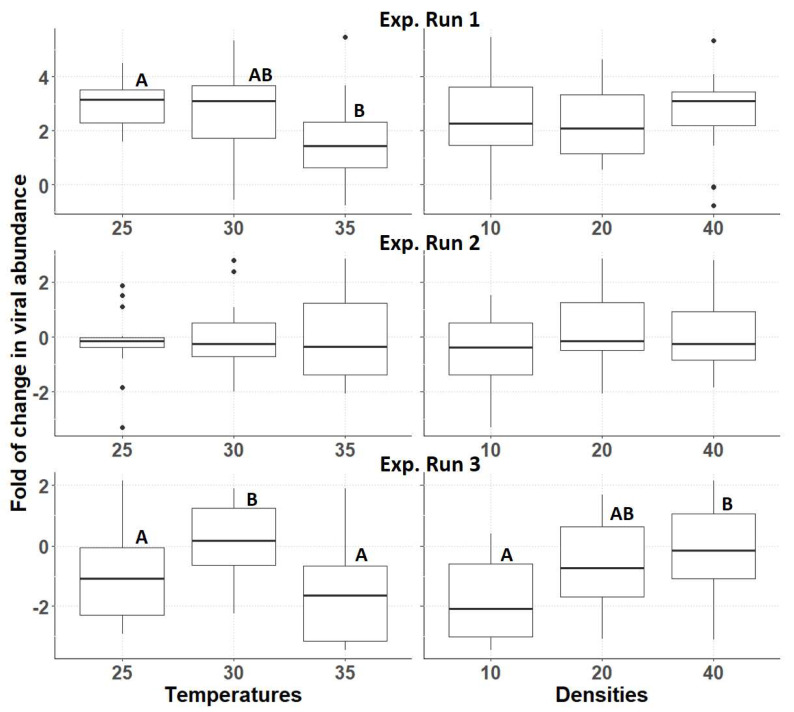
Relative viral abundance in the three experimental runs. The Y-axis represents the fold of change in viral abundance, relative to the start levels, while the X-axis represents the different temperatures (three left−hand panels) and different densities (three right-hand panels). The boxplots show the lower and upper quartiles; vertical lines extend to minimum and maximum values, black dots represent outliers. Different letters indicate significant differences between the treatments.

**Table 1 insects-14-00588-t001:** Primers used in qPCR (EF1- Elongation factor 1, and AdDV96- Acheta domesticus densovirus).

Name	Reference	Primer Sequence 5′ to 3′ (Forward/Reverse)	Amplicon Size (bp)	Gene Target
EF1	This study	GGAAATCAAGAAGGAAGTCAGC/GGCATCCAAAGCCTCAATAAG	199	*A. domesticus* elongation factor 1 alpha gene
AdDV 96	Duffield et al. [18]	GCGAGCAATCCCGACTACTA/CGCGTTGTTGATGTCCTTCC	96	Non-structural protein

## Data Availability

The data is available upon request.

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
