# Peer review of "Effects of Temperature and Density on House Cricket Survival and Growth and on the Prevalence of Acheta Domesticus Densovirus"

_insects, 2023, doi:10.3390/insects14070588_

Round 1
Reviewer 1 Report
Authors investigated the independent and combined effects of temperature (25, 30, and 35C) and population density (10, 20, and 40 individuals) on survival, biomass, and viral abundance of a high pathogenic densovirus in Acheta domesticus house crickets. For survival, they found no effect of density, but an effect of temperature where survival decreased from 25 – 30C (but was not different at 35C). Total biomass was unsurprisingly impacted by initial density with a significant interaction with density and temperature. The overall trend was that temperature increased biomass within density treatments in all but the 40-density treatment where biomass was the same. For viral abundance, they had a large batch effect and so chose to analyze data for each experimental run independently and found some effects of temperature and density, depending on the run. Overall, I think the question asked is a good one and has the potential to be really informative to producers. However, I think the design and results fall short of being useful for researchers or producers for several reasons.
Here are my main concerns:
· Results section 3.2.1: It is unclear to my why authors measured total biomass instead of individual body size. Because population density is varied in this experiment, the measurement of biomass across these treatments is not that interesting (obviously you would find an increase in total biomass as density increases). I would argue what’s more interesting here is the effect of mass per capita by measuring individual body mass. I think they should be able to calculate this number given what they measured (total biomass / number of individuals) although individual measurements would be preferred (e.g., to determine the variance in body size within treatments).
· It’s obviously unfeasible to measure effects of density at a production scale for this group, but the choice of treatment population densities here (10, 20, and 40 crickets per box) seems quite small. Just for standard small lab colonies (not even at production scale), it’s common to keep 300-500 adults in a ~40 L box at a time. There are many previously published papers investigating effects of population density in crickets, so what I am looking for some justification for their choices of density treatments. I would struggle to see how these results could be generalized to any meaningful production approaches.
· The authors target the Acheta domesticus elongation factor 1 alpha gene as the host reference gene using primers they designed but make no mention of how it performs (i.e., report primer efficiency or stability across groups). Moreover, they would have ideally assessed at least two genes to compare how they perform. This is necessary for previously unpublished reference genes.
· I am curious about the batch effect for viral abundance (lines 233 – 235). Could the authors make any guesses as to why this might be? I have heard of a seasonal effect on disease outbreaks where warmer and wetter temperatures correlate with an increase in viral load and am wondering if this could explain this anomaly. What was the timing between batches? Additionally, was all molecular work conducted at the same time as their detection methods are quite sensitive?
Minor:
Lines 89-91: There are several citations for data on growth measures in Acheta domesticus.
Information on the AdDNV (or related densoviruses) in the introduction is quite sparse. What do we know about transmission and prevalence?
It is becoming increasingly apparent that crickets can harbor multiple pathogenic viruses. What these authors know about the standing viral diversity in their populations? Further, what do the authors know about viral dynamics of AdDNV in their standing cricket populations? Do their crickets experience active viral outbreaks? Were the crickets chosen suffering from an active outbreak?
What is the approximate density of crickets in source rearing boxes?
Author Response
Dear Reviewer,
Thank you for your feedback and suggestions on our manuscript. We have revised the text and referred to the changes made with the page and line numbers. We hope the changes implemented and the answers to your questions will be sufficient.
Best Regards
Jozsef Takacs

Reviewer 2 Report
I will suggets to increase the results but the data are really interesting. They suggest the temperature could reduce the outbreak of AdDV.

Author Response

(The authors gave the same response as above.)

Round 2
Reviewer 1 Report
I appreciate the author's revisions. I still have reservations with the small population sizes used in their density treatments (I think they are far too small to be useful in field or lab applications) and I think their rational is weak. For example, they do not take into consideration what the total number of available conspecifics might mean for AdDNV transmission or behavior (e.g., cannibalism) irrespective of surface area or food availability. Obviously this cannot easily be changed without rerunning the entire experiment, but this limitation could be noted in their discussion with citations for standard production scale. I think this is especially important as the authors have framed this paper with a focus on improving mass rearing practices. I also stand by my argument that the average individual body mass should be presented over total biomass as the authors are directly influencing total biomass via population density treatments (unsurprisingly, the largest populations had the heaviest total biomass).
Author Response
Dear Reviewer,
We have updated the manuscript, which now includes in the results section two new sub-sections, which are discussing the the individual weights of the female and male crickets.
In lines 121-125 of the Matterials and methods we have updated the text discussing the densities tested and the reasoning why those were choosen. In the Discussion further editing was made to the text between lines 348-354 regarding the densities tested and how those can be extrapolated.
We are gratefull for your input and recommendations and hope the updated manuscript will meet all the required changes.
Best regards
The authors